# Synthesis and DNase I Inhibitory Properties of New Squaramides

**DOI:** 10.3390/molecules28020538

**Published:** 2023-01-05

**Authors:** Nina Ruseva, Hristina Sbirkova-Dimitrova, Mariyana Atanasova, Ana Marković, Žaklina Šmelcerović, Andrija Šmelcerović, Adriana Bakalova, Emiliya Cherneva

**Affiliations:** 1Department of Chemistry, Faculty of Pharmacy, Medical University of Sofia, 2 Dunav Str., 1000 Sofia, Bulgaria; 2Institute of Mineralogy and Crystallography “Akad. Ivan Kostov”, Bulgarian Academy of Sciences, Acad. G. Bonchev Bl. 107, 1113 Sofia, Bulgaria; 3Department of Pharmacy, Faculty of Medicine, University of Niš, Bulevar Zorana Đinđića 81, 18000 Niš, Serbia; 4Center for Biomedicinal Science, Faculty of Medicine, University of Niš, Bulevar Zorana Đinđića 81, 18000 Niš, Serbia; 5Department of Chemistry, Faculty of Medicine, University of Niš, Bulevar Zorana Đinđića 81, 18000 Niš, Serbia; 6Institute of Organic Chemistry with Centre of Phytochemistry, Bulgarian Academy of Sciences, Acad. G. Bonchev Str., Build. 9, 1113 Sofia, Bulgaria

**Keywords:** monosquaramides, DNase I inhibition, XO inhibition, docking studies

## Abstract

Three new monosquaramides (**3a**–**c**) were synthesized, characterized by IR, NMR and X-ray, and evaluated for inhibitory activity against deoxyribonuclease I (DNase I) and xanthine oxidase (XO) in vitro. The target compounds inhibited DNase I with IC_50_ values below 100 μM, being at the same time more potent DNase I inhibitors than crystal violet, used as a positive control. 3-Ethoxy-4-((1-(pyridin-3-yl)propan-2-yl)amino)cyclobut-3-ene-1,2-dione (**3c**) stood out as the most potent compound, exhibiting a slightly better IC_50_ value (48.04 ± 7.98 μM) compared to the other two compounds. In order to analyze potential binding sites for the studied compounds with DNase I, a molecular docking study was performed. Compounds **3a**–**c** are among the most potent small organic DNase I inhibitors tested to date.

## 1. Introduction

Squaramides (cyclobutenedione amine derivatives) are a special class of carboxylic acid amines considered as vinylogous amides [1,2]. Over the past decades, this intriguing class of compounds has attracted the attention of many scientists. This great deal of interest is due to their unique properties as a result of the chemical structure: the rigid and aromatic character of the ring, and the probability to form four strong hydrogen bonds: two hydrogen-bond acceptors and two hydrogen-bond donors. Compounds containing squaramide fragments have a diverse range of applications, including materials chemistry [2,3,4], organic catalysis [5], chemical biology [6], squaraine dyes [7,8], anion-sensing polymers [9,10], molecular recognition and medicinal chemistry (Figure 1) [6].

Squaric acid and some squaramide-containing compounds have entered clinical trials. Navarixin (MK-7123) is in phase 2 clinical trials for the treatment of chronic obstructive pulmonary disease [11,12]. Perzinfotel (EAA-090), another drug in phase 2 clinical trials, finds application as a potent NMDA antagonist and for the treatment of neuropathic pain [6,13,14]. For the treatment of warts, squaric acid is used, and the dibutyl ester of squaric acid is known as a drug for alopecia [15].

Deoxyribonucleic acid (DNA) is one of the key macromolecules necessary for the continuity of life. Deoxyribonuclease I (DNase I) is one of the most important enzymes in the human body that catalyzes the hydrolysis of DNA producing 5′-oligonucleotides. This enzyme is one of the main nucleases responsible for DNA fragmentation during programmed cell death [16,17]. Elevated levels of DNase I can lead to increased DNA fragmentation, excessive cell death and the development of numerous pathophysiological conditions, including neurodegenerative, cardiovascular and autoimmune, among others [16,18]. A significantly increased DNase I activity was found in the brains of patients with Alzheimer’s disease, which indicates the role of this enzyme in the apoptotic process known to occur in this disease [19]. Therefore, DNA inhibition represents an important mechanism of DNA protection against premature degradation, and DNase I inhibitors may be of exceptional therapeutic importance.

Xanthine oxidase (XO) is a molybdoflavoprotein, highly active in the liver and intestine of the human organism. This enzyme participates in purine metabolism by catalyzing the oxidative hydroxylation of hypoxanthine to xanthine, and xanthine to uric acid. In this reaction, reactive oxygen species, including superoxide anion radicals, hydrogen peroxide and hydroxyl radicals, are also generated. The production of uric acid and reactive oxygen species can lead to gout and oxidative tissue damage [20,21]. XO inhibitors can therefore be of great importance in the prevention/therapy of hyperuricemia and gout.

Adding to the above the fact that the number of known organic DNase I inhibitors is relatively small, that there is no DNase I inhibitor defined as a “golden standard”, as well as the fact that the therapeutic use of approved XO inhibitors (allopurinol and febuxostat) is associated with the possibility of developing serious side effects, there is a need to find new effective DNase I and XO inhibitors, especially those with potential therapeutic applications.

Herein we present the synthesis and identification of new monosquaramides containing pyridine moiety. Synthesized compounds were evaluated in vitro for DNase I and XO inhibitory properties. Finally, molecular docking and molecular dynamics simulations were performed to elucidate structural features necessary for their properties.

## 2. Results and Discussion

### 2.1. Chemistry

Three monosquaramides (**3a–c**) bearing a pyridine moiety were prepared via a condensation reaction between commercially available diethyl ester of squaric acid (**1**) and derivatives of pyridine (**2**) by Tietze’s synthesis [22] (Figure 1). The synthesized compounds were isolated with good chemical yields and high purity. They were stable in air, either as a solid or in solution, as expected. The identification of the target compounds was achieved by using the FT-IR, NMR and single-crystal X-ray diffraction spectroscopic techniques.

### 2.2. Structure Characterization

#### 2.2.1. Spectral Analysis

The IR spectra of the monosquaramides **3a**–**c** showed weak absorption bands in the region 3180–3174 cm^−1^ for N–H stretching vibrations. The C=O stretching vibrations were characterized by a weak symmetric band around 1802 cm^−1^, while the asymmetric band was strong at 1694–1702 cm^−1^. The band for C=C stretching vibrations appeared at 1590–1592 cm^−1^. It was broad due to the overlapped stretching ν(C=C) vibrations of the pyridine ring.

#### 2.2.2. Crystal Structure Analysis

Single-crystal analyses disclose that compounds **3a** and **3c** crystallize in a centrosymmetric manner in monoclinic *P*2_1_*/n* and triclinic *P*-1 space groups, respectively (Figure 2). A summary of the most important crystal data and refinement indicators is provided in Section 4. Materials and Methods. The bond lengths and angles of **3a** and **3c** are comparable to those observed in similar compounds [23] (Appendix A). Compounds **3a** and **3c** are built up by identical squaric acid and pyridine-3-aromatic ring systems (**3b** has a pyridine-4-moiety), and only the part linking the aromatic rings differs (ethylamine vs. isopropylamine). The squaric acid and pyridine aromatic ring systems present in **3a** and **3c** are essentially planar (rmsd in the range 0.001 to 0.0017 Å). The angle between the mean planes of the squaric acid and the pyridine-3 moiety is 41.25°(4) and 46.11°(5) for **3a** and **3c**, respectively, and it shows that the preferred molecular geometry is cis. Interestingly, in the almost identical compound but with pyridine-2-yl (ZEBXAH [24]), the angle between the mean planes of the squarate and pyridine is 7.60°, thus the molecule is almost “linear” (Figure 3). Bearing in mind the similarity (bond lengths and angles, identical aromatic ring systems, potential hydrogen-bond donor and acceptor), one should expect a similar pattern for the hydrogen-bonding interactions in **3a** and **3c**. Indeed, in both structures, classical N-H…N hydrogen-bonding interactions link the molecules (Appendix A). No π…π stacking interactions are detected. However, the three-dimensional packing of the molecules in the crystal is stabilized by a multitude of weak C-H…O interactions (Appendix A). Interestingly, the DFT calculation reported below displays a conformation that is close to the linear one.

#### 2.2.3. Quantum Chemical Modeling

Being unable to grow a single crystal of **3b**, quantum chemical studies were applied to explore the molecular structure using the B3LYP/6-31++G(d) level (Figure 4). The results were compared with the data obtained from the X-ray analysis of **3a** (Table 1). The calculations in the gaseous state showed that the geometric parameters of **3b** are in very good agreement with those found in **3a**’s X-ray studies in solid state. As it can be seen in Figure 4, the pyridine fragment is perpendicularly oriented to the monosquaramide ring.

### 2.3. ADME Screening

The ability to cross the blood–brain barrier and gastrointestinal absorption are two pharmacokinetic characteristics that play a key role in drug discovery. The ADME profile of the target compounds was estimated in silico using free SwissADME web tools [25].

The consensus log P, calculated as an average of the five available methods for logP prediction [25] of the studied inhibitors, vary in the interval between 1.02 and 1.35. These values indicate that the compounds are slightly lipophilic, and their hydrophilic–lipophilic properties are very close to the well-balanced character needed for a good (fast and full) absorption of a drug that is maintained via passive diffusion. Another drug bioavailability indicator for drug bioavailability, blood–brain barrier (BBB) permeability and intestinal absorption is the TPSA (Topological Polar Surface Area) [26]. The predicted TPSA values of the study compounds are 68.29 Å2, indicating a good intestinal absorption, and no BBB penetration as the value is greater than 60 Å2. Additionally, these results are confirmed by using the BOILED-Egg (Brain Or Intestinal EstimateD permeation) method for predicting gastrointestinal absorption (GI) and BBB permeability, based on the lipophilicity and polarity of small molecules [27] implemented in SwissADME [25]. The oral bioavailability of the drugs might be influenced by molecular flexibility [26]. Six rotatable bonds are predicted for monosquaramides **3a–c**. The compounds show good hydrogen-bonding potential: 1 H-bond donor and 4 H-bond acceptors. All monosquaramides fulfil the five drug-likeness filters, i.e., the rules of Lipinski, Ghose, Veber, Egan and Muegge, implemented in SwissADME [25]. Based on the calculated ADME parameters, the target compounds should have an excellent gastrointestinal absorption, which is a very favorable characteristic for oral bioavailability. They are not predicted as P-glycoprotein substrates. All the compounds are expected to inhibit the CYP1A2 isoform, and compound **3c** can inhibit CYP2C19 as well. This inhibition is a potentially adverse characteristic, because it could lead to toxicity or lack of drug efficacy [28,29]. According to the calculated Log Kp values, the skin permeation ability of the synthesized compounds was expected to be incredibly low. The Log Kp coefficient is −6.73 for **3c**, and −6.95 for **3a–b**. Monosquaramides are soluble or moderately soluble in water, according to the SwissADME server [25].

### 2.4. Biological Evaluation

#### 2.4.1. Antiproliferative Activity

A preliminary screening of the cytotoxic activity of compounds **3a–c** was evaluated using the MTT assay described by Mossman [30] with some modifications [31]. The method is based on the reduction of the yellow tetrazolium salt MTT [3-(4,5-dimethylthiazol-2-yl)-2,5-diphenyltetrazolium bromide] to a violet formazan via the mitochondrial succinate dehydrogenase in viable cells. The cell lines used were the HL-60, MCF-7 and MDA-MB-23 human tumor cell lines. Unfortunately, none of the compounds tested exhibited in vitro antiproliferative activity on the human tumor cell lines. IC_50_ values were greater than 200 μM.

#### 2.4.2. DNase I and XO inhibition

Compounds **3a**–**c** were included in an in vitro DNase I inhibition assay. As a result, all three monosquaramides inhibited DNase I with IC_50_ values below 100 µM, and were also more potent than crystal violet (IC_50_ = 357.92 ± 40.74 µM), which was used as a positive control in the absence of a “golden standard”. Compound **3c**, having a 1-(pyridin-3-yl)propan-2-yl moiety, stood out as the most potent DNase I inhibitor, having a slightly higher IC_50_ value (48.04 ± 7.98 µM) compared to those of 2-(pyridin-3-yl)ethyl (compound **3a**) and 2-(pyridin-4-yl)ethyl (compound **3b**) analogues (IC_50_ = 54.53 ± 8.07 and 57.81 ± 9.67 µM, respectively). It is important to point out that compounds **3a**–**3c** are among the most potent small organic DNase I inhibitors tested to date [32,33,34,35,36,37,38,39,40,41,42,43].

Compounds **3a**–**c** were included in an in vitro XO inhibition assay. However, none of the three monosquaramides inhibited XO at the concentration tested (IC_50_ > 200 µM).

## 3. Molecular Docking Study

A blind docking covering the entire protein surface of the crystal structure of the bovine pancreatic DNase I (PDB code: 2DNJ) [44] was performed using AutoDock Vina v. 1.2.0 of AutoDock Suite (Center for Computational Structural Biology, CCSB, La Jolla, California, USA) [45]. To identify the binding site (BS) of the studied compounds, 13 distinct protocols with different technical parameters were applied. The top 9 poses were kept for each protocol. Consequently, 117 poses (13 protocols × 9 poses) were available for each compound. Four BSs were identified for the studied compounds (Figure 5). Among the total 351 disposable poses (3 compounds × 117 poses), 12 poses (~3%) were placed at BS3 and 5 poses (~1%) at BS4. Additionally, these BSs were located far from the contact regions between the protein and the DNA. For these reasons, the contributions of BS3 and BS4 were assumed to be negligible and were omitted from consideration. BS1 and BS2 were located in both regions where the substrate and the DNase I formed contacts. The amino acids constructing BS1 are Thr10, Phe11, Gly12, Glu13, Thr14, Lys15, Val40, Arg41, Asp42, Ser43, His44, Glu69, Pro70, Leu71, Ser75, Tyr76, Lys77, Glu78 and Arg79. BS1 accommodated 106 poses (30 %) of all 351 for the studied compounds. Of them, 48 were for **3a**, 28 were for **3b**, and 30 for **3c**. Interestingly, the top poses were derived at this BS. The highest affinity for **3a** and **3c** was estimated to be −6.1 kcal/mol, while for **3b**, it was 5.9 kcal/mol. The three inhibitors shared similar top poses at BS1 (Figure 5). Therefore, the contacts that occurred at BS1 were analogous (Appendix A and Figure 6). Figure 5 depicts the interactions of the most potent inhibitor (**3c**). A hydrogen-bond network stabilized the accommodated ligands at BS1. Two hydrogen bonds were formed between the N atom from the pyridine ring and the H atoms of the side chain OH and the backbone NH groups of Thr14. Another two bonds occurred between the methoxy O atom of the monosquaramide and the H atoms of the backbone amino groups of Asp42 and Ser43. Both carbonyl O atoms took part in a hydrogen-bond network, including the H atom of the backbone NH group of Lys77 and the positively charged guanidino group of Arg41. Additionally, two amide-π stacking contacts were formed between the ligands and the DNase I. One was between the pyridine ring and the amide bond of Phe11. In the second one, the squaramide and amide bond of Arg41 were involved (Appendix A).

The residues Asn7, Arg9, Glu39, Arg41, Gly72, Arg73, Asn74, Ser75, Tyr76, Glu78, Ser110, Arg111, His134, Ser135, Ala136, Pro137, Ser138, Asp168, Asn170, Tyr175, Thr203, Ala204, Thr205, Thr207, Cys209, Tyr211, Asp251 and His252 constituted BS2. This is consistent with similar studies using different docking tools [35,39,43]. A total of 228 (65%) poses out of the 351 were placed at BS2: 67 of **3a**, 84 of **3b**, and 77 of **3c**. BS2 occupied the vast shallow groove where the octanucleotide binds DNase I (Appendix A). Due to its huge surface, the small ligand molecules were able to bind at different subsites of BS2. Consequently, this might be one of the reasons for the highly populated binding region. The affinities of the top poses at this BS for the studied compounds were estimated to be −5.8 kcal/mol, −5.7 kcal/mol and −6.0 kcal/mol for **3a**, **3b** and **3c**, respectively. The three top poses (affinity = −6.0 kcal/mol) of the most potent inhibitor (**3c**) were similar and located at one single subsite (Appendix A and Figure 7). This subsite accommodates some of the top poses of **3a** and **3b** (Appendix A). The intermolecular interactions of **3c** are depicted in Figure 7 and Appendix A. A hydrogen-bond network stabilized the formed complexes. These interactions occurred between the carbinyl O atoms, and Arg73, Asn74 and Tyr76. The pyridine N atom and Arg111 took part in another hydrogen bond. The methoxy O atom was involved in two hydrogen bonds with Ser110 and Arg111. Two π-π stacking contacts were detected between the pyridine and squaramide rings, and Tyr76. One cation-π interaction was formed between the squaramide moiety and Arg111. Additionally, the pyridine ring was involved in one anion-π contact with Glu78.

There are three different top poses of ligand **3a** at BS2 estimated with an affinity of −5.8 kcal/mol. Their interactions are depicted in Appendix A. A number of hydrogen bonds stabilized the complexes formed. They occurred between the carbonyl atoms and Asn7, Arg9, Arg73, Asn74, His134 and His252. The methoxy O atom was involved in two hydrogen bonds with Arg111 and His252. The pyridine and squaramide moieties participated in π-π stacking with Tyr76 and Tyr211 (Appendix A).

Four top poses were identified with an affinity of −5.7 kcal/mol for compound **3b** at BS2. In an analogous manner, the inhibitor was involved in many hydrogen bonds (Figure 6). They occurred between the CO groups and Arg9, Arg73, Asn74, Tyr76, His134 and His252. The methoxy O atom participated in one hydrogen bond with Ser110. The H atom at the NH group at the linker formed three hydrogen bonds with Glu39, Tyr76 and Glu78. The pyridine N atom was involved in one hydrogen bond with Arg111. The poses were additionally stabilized with: one cation-π contact between the pyridine cycle and Arg111; two anion-π interactions (one between the squaramide ring and Glu39 and one between the pyridine and Glu78); and one π-π stacking between the pyridine ring and Tyr76 (Appendix A).

The conducted blind docking revealed that the studied DNase I inhibitors (**3a**, **3b** and **3c**) manifested their activity via two binding sites: BS1 and BS2 (Figure 4). Both sites are located within the two contacting regions between the d(GCGATCGC)2 and the DNase I. BS1 is at the secondary contact region between the protein and the dinucleotide. BS2 is located near the primary active center at the major contact area, which presents a shallow groove shape between the two central β-sheets and side loops.

It has been reported that residues Tyr76 and Arg41 formed contacts in the minor groove of the octanucleotide leading to geometry changes, namely a great distortion of the phosphate backbone conformation, and the base-pair orientation and stacking [43]. Our results showed that both residues were involved in intermolecular interactions between the studied inhibitors and the enzyme. Additionally, the identified secondary binding site, BS1, is consistent with the assumed secondary active site (consisting of Glu13, Arg41, Asp42, Ser43 and His44) in DNase I, which could be activated in specific cases, such as the presence of Mn2+ and other transition ions [44].

The most contact residues at both identified BSs are the same for both the studied compounds and the dinucleotide. These are Arg9, Thr14, Lys15, Arg41, Ser43, Asn74, Tyr76, Ser110, Arg111 and Tyr211 [42]. Additionally, at BS2, the studied inhibitors formed contacts with the most important amino acids (Glu39, Glu78, His134, His252 and Asn170) of the catalytic center [44,46,47,48], and, thus, directly hindered or even inhibited the binding of the DNA.

These findings might be related to the multiple mechanisms of action of the studied inhibitors manifesting via more than one binding site in DNase I. One of the possible mechanisms might be a classic direct competitive binding to the active center and blocking of the substrate access. Another possibility might be that two or more small molecules bind simultaneously at the huge shallow groove where the DNA molecule contacts the enzyme (i.e., in an allosteric manner), to hinder substrate binding to the protein. A third possible mechanism could occur in the secondary binding site, denoted as BS1, located at the second contact region between the enzyme and the octanucleotide, where inhibitors might interfere their interactions and, thus, allosterically manifest their activity. As the ligands are small molecules, all three mechanisms or their combinations might manifest simultaneously for inhibiting DNase I.

## 4. Materials and Methods

**General**. All the chemicals used were of analytical grade. The microchemical analyses (C, H, N) of the compound were carried out with a EuroEA 3000—Single (EuroVectorSpA. Milan, Italy). IR spectra were recorded in the range of 4000–100 cm^−1^ on a Bruker Invenio R spectrophotometer (Bruker, Ettlingen, Germany), ATR (attenuated total reflectance) mode with a diamond crystal accessory. The spectra were referenced to air as a background by accumulating 100 scans, at a resolution of 2 cm^−1^. The ^1^H NMR spectrum was recorded on an Avance Neo 400 (400 MHz) spectrometer (Bruker, Ettlingen, GermanyCorrected melting points were determined on a Buchi 540 apparatus (BUCHI, Flawil, Switzerland).

**Synthesis of 3a–c**. Compounds **3a–c** were synthesized according to Tietze’s method [22]. To a solution of corresponding pyridine containing compounds (0.001 mol), diethyl ester of squaric acid (0.001 mol) and excess of triethylamine were added. The reaction mixtures were stirred for 4–7 days and left in refrigerator for 7–10 days. The white- and yellow-colored crystals of **3a** and **3c**, respectively, were obtained.

**3a** Yield: ca. 44% m.p.: 139.8–141.2 °C. Color: light yellow. Elemental Analysis: Calc. for: C, 63.40; H, 5.73; N, 11.38; Found: C, 63.82; H, 5.87; N, 11.69; IR (ATR, cm^−1^): 3175, 1804, 1694, 1590. ^1^H-NMR (400 MHz, DMSO-*d*_6_, δ, ppm): 8.85 (s, 1H, NH); 8.42 (s, 2H, Py); 7.63 (t, 1H, Py); 7-34–7.32 (m, 1H, Py) 4.64–4-52 (m, 2H, CH_2_O); 3.76–3.71 (q, 1H, CH_2_); 3.54–3.52 q, 1H, CH_2_); 2.85–2.84 (t, 2H, CH_2_); 1.35–1.30(m, 3H, CH_3_); ^13^C-NMR (162 MHz, DMSO-*d*_6_, δ, ppm): 176.96; 173.08; 150.52; 148.14; 136.97; 134.29; 123.95; 69.27; 45.30; 44.78; 34.05; 33.70; 15.99.

**3b** Yield: ca. 55% m.p.: 149 °C. Color: white; Elemental Analysis: Calc. for C_13_H_14_N_2_O_3_: C, 63.40; N, 11.38; H, 5.73. Found: C, 63.95; N, 11.27; H, 5.78. IR (ATR, cm^−1^): 3179, 1802, 1702, 1592; ^1^H-NMR (400 MHz, DMSO-*d*_6_, δ, ppm): 8.87 (s, 1H, NH); 8.61 (s, 2H, Py); 7.30 (s, 2H, Py); 4.66 (m, 2H, CH_2_O); 3.82–3.79 (q 1H, CH_2_); 3.65–3.63 (q, 1H, CH_2_); 2,90 (t, 2H, CH_2_); 1.40–1.39 (m, 3H, CH_3_); ^13^C-NMR (162MHz, DMSO-*d*_6_, δ, ppm): 177.06; 173.08; 149.96; 147.76; 137.05; 125.02; 69.29; 44.64; 44.10; 36.02; 35.78; 16.04.

**3c** Yield: ca. 26% m.p.: 153–154.8 °C. Color: white; Elemental Analysis: Calc. for: C, 64.60; H, 6.20; N, 10.76; Found: C, 64.91; H, 6.35; N, 10.53; IR (ATR, cm^−1^): 3345, 3202, 1767, 1707, 629. ^1^H-NMR (400 MHz, DMSO-*d*_6_, δ, ppm): 8.81–8.79 (*d*, 1H, NH); 8.40 (s, 2H, Py); 7.62–7.58 (m, 1H, Py); 7.32–7.29 (m, 1H, Py); 4.62–4.48 (m, 2H, CH_2_O); 4.54–4.48 (m, 1H, CH_2_); 3.34–2.68 (m, 2H, CH_2_); 1.36–1.26 (m, 6H, CH_3_); ^13^C-NMR (162 MHz, DMSO-*d*_6_, δ, ppm): 189.40; 182.27; 176.45; 172.15; 150.83; 148.12; 137.24; 123.74; 69.18; 52.76; 51.94; 21.64; 21.22; 16.05; 15.97.

**Computational details**. All theoretical calculations were performed using the *Gaussian 03, Revision C.02*; Gaussian Inc.: Wallingford, CT, USA, 2004 [49]. Optimization of the structures of target compounds was carried out by hybrid DFT calculations, employing the B3LYP (Becke’s three-parameter non-local exchange) [50,51] functional correlation, 6-311++G(g,p) basis set.

**Single-crystal X-ray diffraction (SCXRD) studies**. Single crystals with suitable quality of **3a** and **3c** were fished on a nylon loop. The diffraction data for both compounds were collected on a Bruker D8 Venture diffractometer equipped with a PhotonII CMOS detector using micro-focus MoKα radiation (λ = 0.71073 Å). Data reduction was performed with APEX4 software [52]. The structures were solved with intrinsic methods using ShelxT [53] and refined by the full-matrix least-squares method on *F*^2^ with ShelxL program [53]. All non-hydrogen atoms were located successfully from Fourier map and were refined anisotropically. Hydrogen atoms were placed on calculated positions (C–H_methyl_ = 0.96 Å, C–H_aromatic_ = 0.93 Å and C–H_methylenic_ = 0.97 Å) riding on the parent atom (Ueq = 1.2). Most important data collection and crystallographic refinement parameters for **3a** and **3c** are provided in Table 2. Complete crystallographic data for the structure of compounds **3a** and **3c** reported in this paper have been deposited in CIF format with the Cambridge Crystallographic Data Center as 2,213,349 and 2,213,350. These data can be obtained free of charge via http://www.ccdc.cam.ac.uk/conts/retrieving.html (accessed on 1 January 2020). (or from the CCDC, 12 Union Road, Cambridge CB2 1EZ, UK; Fax: +44-1223336033; E-mail: deposit@ccdc.cam.ac.uk).

**Biological evaluation**. The present study describes a preliminary screening of the cytotoxic effect of the compounds **3a–c** on three human tumor cell lines: HL-60, MCF-7 and MDA-MB-231. The survival fraction was calculated as the percentage of the untreated control. The experimental data were processed using GraphPad Prism software and were fitted to sigmoidal concentration/response.

**Evaluation of DNase I inhibition**. In vitro determination of the inhibitory activity of compounds **3a–c** against bovine pancreatic DNase I was performed using the spectrophotometric method described by Kolarević et al. [32], which is based on measuring the absorbance of the formed oligonucleotides at 260 nm.

**Evaluation of XO inhibition**. In vitro determination of the inhibitory activity of compounds **3a–c** against XO was performed using the spectrophotometric method described by Smelcerovic et al. [54], which is based on measuring the absorbance of the formed uric acid at 293 nm.

**ADME Screening.** ADME calculations were performed using the free web tool SwissADME of the Swiss Institute of Bioinformatics (https://www.sib.swiss, accessed on 20 April 2022). The tool is based on binary classification, multiple linear regression and support vector machine algorithms performed over large data sets of known inhibitors/non-inhibitors, as well as on substrates/non-substrates [55,56].

**Molecular docking**. The crystal structures of **3a** and **3c**, as well as the theoretically optimized structure of **3b**, were used for molecular docking calculations on the crystal structure of bovine pancreatic DNase I in complex with dinucleotide d(GCGATCGC)2 (PDB ID: 2DNJ, Resolution 2 Å) [44] via AutoDock Vina v.1.2.0 [45]. The dinucleotide and water molecules were removed. The binding site (BS) on DNase I is currently unknown. Therefore, a blind docking search covering the whole protein surface was applied with the following grid-box parameters: x center: 52.369 Å, y center: 28.297 Å, z center: 35.826 Å; number of points in x-dimension: 48, y-dimension 48, z-dimension: 50. The calculations were performed at rigid proteins and flexible inhibitors. The exhaustiveness and energy range values were varied to provide a deep conformational search on the whole protein surface. Their intervals were from 8 to 50 for the exhaustiveness, and from 3 to 15 for the energy range.

## 5. Conclusions

Three pyridine-containing monosquaramides as potent DNase I inhibitors were synthesized and characterized. Two of the compounds, **3a** and **3c**, crystallize centrally-symmetrically in monoclinic space group P21/n and ternary space group P-1, respectively. The ADME screening predicted that the compounds had suitable bioavailability and high levels of gastrointestinal absorption. The molecular docking calculations revealed the potential binding sites on the DNase I surface where the newly synthesized compounds accomplished their activity. All three monosquaramides inhibited DNase I with quite similar IC_50_ values (ranging from 48.04 to 57.81 µM), but none inhibited XO (IC_50_ > 200 µM).

These structures represent a good basis for further in vivo studies, as well as for the design and synthesis of new and even more effective enzyme inhibitors.

## Data Availability

Not applicable.

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
