# Peer review of "Synthesis and DNase I Inhibitory Properties of New Squaramides"

_molecules, 2023, doi:10.3390/molecules28020538_

Round 1

Reviewer 1 Report

The present submission describes an integrated experimental and theoretical study on the interaction of new squaramides and the enzyme DNase I. The inhibition of DNase I may be relevant in the control of diseases related to excessive apoptosis; therefore, this type of investigation is welcome. Three new compounds were synthesized and characterized, two by X-ray crystallography. The 3D structure for the third derivative was proposed by DFT calculation.  The compounds were not active in cells, but inhibited DNase I with IC50 < 100 microM. From the docking simulation, the two main binding sites were identified as BS1 and BS2, both located in the DNA binding region. The best poses were in BS1 and the highest affinity was -6.1 for 3c, which was also the most potent derivative. Overall, the docking procedure was interesting and can be considered "good practice" for similar studies where the binding site is not well defined or allosteric binding sites are of interest.

I have few comments and suggestions before final acceptance of the manuscript for publication in Molecules.

1) According to the authors, DNase I “catalyzes the hydrolysis of DNA”, whose process is related to cell apoptosis. However, the compounds were tested against tumor cell lines. I suppose that antitumor activity must go through apoptosis at the end which, on the other hand, is suppressed by DNase I inhibition. The authors could discuss this point in section 2.4.

2) Section 2.4.3 should be removed as the compounds did not show activity against XO.

3) The ‘Computational details’ section includes information for Pt basis, which atoms is not part of the molecules. Please, fix that.

Author Response

We would like to thank the Reviewer for their constructive and valuable comments that can help us to improve the quality of our manuscript. Below we try to respond the Reviewers’ remarks and clear them step by step. We hope that upon the modifications, explanations and corrections that we have made our paper will become suitable for publication in journal Molecules. The answers to the comments are in the same order as Reviewer remarks. All implemented changes are marked with a track changes in the manuscript.

Suggestion # 1.1: According to the authors, DNase I “catalyzes the hydrolysis of DNA”, whose process is related to cell apoptosis. However, the compounds were tested against tumor cell lines. I suppose that antitumor activity must go through apoptosis at the end which, on the other hand, is suppressed by DNase I inhibition. The authors could discuss this point in section 2.4.

Our results indicate that tested compounds 3a-c have the ability to inhibit DNase I activity and thus, due to the involvement of this enzyme in DNA fragmentation during apoptosis, might have potential therapeutic applications in disorders caused by excessive cell death. We agree that DNase I inhibitors suppress the antitumor activity, but we still wanted to perform this assay because DNase I-induced apoptosis represents only one of the cell death mechanisms. In addition to apoptosis, which has been shown to play a key role in killing tumor cells in response to cytotoxic agents, there are also some other morphological types of cell death (autophagy, necrosis...) (Lossi L. Biochem J, 2022, 479:357-384). There have been findings that cell death of HL-60, MCF-7 and MDA-MB-231, tumor cell lines tested in this study, can rely not only on apoptosis (type I programmed cell death) but also on autophagy (type II programmed cell death). Our results showed that none of the tested compounds exhibited antiproliferative activity.

Suggestion # 1.2:  Section 2.4.3 should be removed as the compounds did not show activity against XO.

It has been done.

Suggestion # 1.3: The ‘Computational details’ section includes information for Pt basis, which atoms is not part of the molecules. Please, fix that.

It has been corrected.

Reviewer 2 Report

As indicated in the manuscript, compounds 3a-c have been characterized by NMR, however only the data for 3b are indicated, the other two compounds should also be included, since they were not found in the supplementary material.

If possible cif archives should be deposited at CCDC or at least in the suplementary material in order to giver a more complete information for readers.

Author Response

We would like to thank the Reviewer for their constructive and valuable comments that can help us to improve the quality of our manuscript. Below we try to respond the Reviewers’ remarks and clear them step by step. We hope that upon the modifications, explanations and corrections that we have made our paper will become suitable for publication in journal Molecules. The answers to the comments are in the same order as Reviewer remarks. All implemented changes are marked with a track changes in the manuscript.

Suggestion # 2.1:  As indicated in the manuscript, compounds 3a-c have been characterized by NMR, however only the data for 3b are indicated, the other two compounds should also be included, since they were not found in the supplementary material.

It has been done.

Suggestion # 2.2: If possible cif archives should be deposited at CCDC or at least in the suplementary material in order to giver a more complete information for readers.

Structures have been deposited at CCDC with numbers 2213349 and 2213350. the last are also included in Table 2.

Reviewer 3 Report

In the manuscript entitled “Synthesis and DNase I inhibitory properties of new squaramides”, the authors described the synthesis, characterization, and in vitro evaluation of inhibitory activity against deoxyribonuclease I (DNase I) and xanthine oxidase (XO). The study deserves to be published at your esteem journal, but there are some comments regarding the manuscript. Authors may find useful to consider following comments and suggestions in preparation of the manuscript. Nevertheless, I believe the paper can be accepted for publication after major revision.

Some comments and corrections for authors:

1.     Some minor, grammatical, or typological errors must be corrected.

2.     In introduction section, there should be a figure that clearly showing the importance of the biologically important squaramides, if available.

3.     The compounds 3a and 3c have no any data related with NMR analysis along with MS analysis. The authors must characterize the structure fully by using proton and carbon NMR and mass analysis.

4.     The “ADME” parameters must be added to the mns in details.

5.     Thoroughly check the consistency of references.

Author Response

Suggestion # 3.1: In introduction section, there should be a figure that clearly showing the importance of the biologically important squaramides, if available.

It has been done.

Suggestion # 3.2: The compounds 3a and 3c have no any data related with NMR analysis along with MS analysis. The authors must characterize the structure fully by using proton and carbon NMR and mass analysis.

It has been done.

Suggestion # 3.3: The “ADME” parameters must be added to the mns in details.

It has been done.

Suggestion # 3.4:  Thoroughly check the consistency of references.

It has been done.

Round 2

Reviewer 2 Report

once the changes were made, I think the manuscript is fine to publish

Reviewer 3 Report

The revised version can be accepted for publication.